# How did Ontario healthcare institutions implement and legitimize Covid-19 vaccine mandates? A qualitative multi-method study protocol

**Claudia Chaufan** [ID]*

School of Health Policy & Management, York University, Toronto, Canada

* cchaufan@yorku.ca

## Abstract

### Background

Upon the WHO declaration of Covid-19 as a pandemic, healthcare workers (HCWs) – unvaccinated by necessity – were celebrated as "heroes" for their service under difficult conditions. Later, as some of them resisted vaccine mandates, they were reframed as "threats", regardless of personal behaviour, workplace setting, or evidence of their harmfulness. This discursive shift and the institutional mechanisms supporting it remain underexamined.

### Goal

This protocol outlines a qualitative multi-method study investigating the implementation of Covid-19 vaccine mandates in medical establishments across Ontario, Canada.

### Methods

This qualitative multi-method planned study includes two phases. Phase 1 is an environmental scan of institutional vaccine mandate policies across a purposive sample of diverse Ontario medical establishments. It will track policy implementation timelines, mandates scope, exemptions eligibility criteria, and supporting scientific evidence presented. Phase 2 is a critical interpretive analysis of documents collected in Phase 1 that draws on Max Weber's theory of bureaucracy and legitimacy, Carol Bacchi's "What Is the Problem Represented to Be" (WPR) approach, and Brian Martin's framework on suppression of dissent to identify patterns in the institutional framing and treatment of challenges to mandated vaccination.

**Data availability statement:** This submission reports a study protocol. Data collection has not begun and no data is reported.

**Funding:** The author(s) received no specific funding for this work.

**Competing interests:** The authors have declared that no competing interests exist.

## Expected outcomes

The study outlined in this protocol is expected to yield descriptive and interpretive insights into how bureaucratic structures shaped mandate enforcement and dissent suppression. Results are expected to inform academic debates on institutional legitimacy, governance, and public health ethics.

## Introduction

The planned study outlined in this protocol is motivated by a paradox that began to unfold in the early months of the Covid-19 policy response: healthcare workers (HCWs), unvaccinated by necessity, were celebrated as heroes for their willingness to serve under crisis conditions (see for example [1,2]). Yet many of these same workers would be later reframed as threats to public health – regardless of their protective behaviours, workplace settings, or evidence of their harmfulness (see for example [3,4]).

Notably, policies mandating influenza vaccination for healthcare workers in Canada have historically operated with alternatives—"vaccinate-or-mask," temporary work restrictions during outbreaks, or encouragement rather than compulsion—and have been adjudicated through labour arbitration that stresses least-intrusive means and a balancing of employee rights with patient safety [5]. In parallel, scholarly and professional debate has long been divided on whether influenza mandates are warranted [6], with critics noting their limited high-level evidence for patient benefit and warning against employment-threatening policies, and supporters arguing that mandates can raise staff vaccination and are ethically justified; even popular polls in clinical forums have reflected this split [7].

A similar flexibility appeared to characterize early Covid-era policy—at least on paper. In Canada, for example, Directive 6, issued by the Chief Medical Officer of Health for the province of Ontario, required every covered organization to implement a Covid-19 immunization policy, but preserved institutional discretion in how to operationalize it. The Directive allowed for compliance via vaccination, documented medical exemption, or participation in an educational program with regular antigen testing for those not fully vaccinated [8].

Against this backdrop, the shift from "hero" to "threat" and employer's policy choices raise critical scientific, policy, and ethical questions. Why were vaccine mandates implemented as condition of employment on HCWs who had already faced sustained exposure to Covid-19, continued to care for patients without interruption, and were not shown to pose a demonstrable risk—well before vaccines were even available? What evidence and arguments were presented to support the contention that vaccination would prevent viral transmission in healthcare settings, the key rationale for their implementation? Why was the most extreme vaccination policy chosen by most establishments? How did political and institutional dynamics shape the justifications and enforcement of mandates? And how were dissent, resistance, or requests for accommodation framed within policy discourse?

To address these and related questions, this qualitative multi-method study planned study will investigate the implementation of Covid-19 vaccine mandates in medical establishments in Ontario, Canada. While many studies have focused on what has been framed as "vaccine hesitancy", we focus instead on mandate legitimation and dissent suppression as institutional processes. This examination will document the timeline of mandate introductions, appraise how institutions interpreted and operationalized provincial directives, and identify what scientific, administrative, or rhetorical justifications were used to support their adoption. The work will be grounded in a critical analysis of the social and institutional narratives that shaped HCWs' changing roles throughout the Covid policy response. Instances of such narratives appeared following the policy rollout, when the "hero" label was deployed as a technique of governance and a tool to enforce professional compliance. As Mohammed et al have argued, the language of heroism – replete with militaristic and sacrificial imagery – elevated frontline HCWs while masking serious lapses in workplace safety, labour protections, and staffing shortages [9]. Similarly, Boulton et al. have suggested that public rituals such as balcony clapping and military flyovers further served as forms of "performative allyship" that, while seemingly supportive, helped to normalize institutional neglect by recasting structural vulnerability as moral duty [10]. These worthy critiques notwithstanding, to our knowledge no research has been conducted on the rhetorical and institutional reframing of dissenting HCWs from lauded protectors of public health to perceived threats to it.

We observe that since the launch of the global vaccination campaign, much of the literature that addresses HCWs' concerns with or refusal of vaccination has tended to pathologize dissent, focusing on behavioural correction through education or coercion rather than engaging with the scientific, political, or ethical dimensions of resistance (see for example [11,12]). Rarely considered are the moral and psychological tolls that these policies have imposed – particularly on women, who already face higher rates of burnout, moral distress, and suicide [13,14] nor are their empirical grounds scrutinized. Finally, even less explored is the institutional silence surrounding the demonization of dissenting HCWs, some of whom have faced termination, ridiculing, or blacklisting – or have even witnessed hostility towards unvaccinated patients [15]. These gaps in the literature suggest not only a lack of scholarly curiosity but also a tendency to frame resistance as pathology rather than as a potentially legitimate professional – scientific and ethical – stance, thus our study.

To investigate these issues, we have adopted a qualitative multi-method study design structured in two complementary phases. In Phase 1, we will conduct an environmental scan that involves the systematic collection and document analysis of a broad range of publicly accessible materials - including hospital policies and internal documents, press releases, media coverage, and legal materials such as grievances and arbitration rulings retrieved from the Canadian Legal Information Institute (CanLII). This phase aims to document how individual Ontario healthcare institutions implemented Covid19 vaccine mandates, what options they selected under Directive 6, how choices were justified, and how variation in implementation may reflect broader institutional logics. In Phase 2, we will conduct a critical interpretive document analysis of the materials collected in Phase 1 to examine how dissenting healthcare workers were framed within institutional discourse. Drawing on Max Weber's theory of bureaucracy and legitimacy, Carol Bacchi's WPR framework, and Brian Martin's typology of dissent suppression, this phase explores how institutional actors framed noncompliance, justified their policies, and responded to internal or public contestation. Taken together, the two phases will generate both descriptive and interpretive insights into the bureaucratic, rhetorical, and ethical dimensions of mandate enforcement and dissent suppression in Ontario's healthcare sector.

The planned study builds on a body of work by the lead author and collaborators, including critical analyses in the academic literature of problem representations of HCWs' dissent [16], surveys on the impacts of Covid-19 policy on HCWs' well-being and patient care [17,18], thematic analyses of HCWs lived experience [15], and the medicalization of dissent with official Covid policies [19,20]. It is part of two larger projects registered in the Open Science Framework that examine the impact of Covid-19 policies on healthcare work and governance more broadly (https://osf.io/z5tkp; https://osf.io/84kbr/). By examining the tensions in the healthcare workplace through the lens of policy implementation, institutional authority, and discourse, the study aims to illuminate a neglected dimension of the Covid-19 policy response: how policies

framed as protective may have functioned as instruments of exclusion, control, or compliance enforcement – especially against those within the healthcare system who raised difficult, but necessary, questions.

## Theoretical framework

The interpretive phase of this planned study is grounded in the sociology of Max Weber, particularly his conceptualizations of bureaucracy, rational-legal authority, and ideal types. These concepts will help to illuminate how healthcare institutions implemented Covid-19 vaccine mandates, legitimized those decisions, and framed or managed internal dissent within their bureaucratic structures. Weber's model of bureaucracy emphasizes rule-governed conduct, hierarchical organization, and a clear distinction between personal and official roles. These features were especially pronounced at the launch of the global vaccination campaign, as healthcare institutions invoked procedural authority to justify exclusionary mandates. That said, and as Byrkjeflot has argued [21], Weber's model has often been misinterpreted as primarily about efficiency and rule-governed behaviour in the context of hierarchical institutions, when it could be as much or better understood as a theory of legitimacy – an insight central to this study's analysis of institutional behaviour.

To organize institutional variation in mandate implementation against the range of possible policy options afforded by Directive 6, the planned study will also apply Weber's concept of "ideal type" [22], an analytical construct that emphasizes certain features of a phenomenon in order to enable comparison. Ideal types will be used to develop heuristic categories of institutional response, ranging from minimalist compliance with provincial guidance to maximalist exclusion of unvaccinated staff. These categories are not meant to capture reality perfectly, but to provide clarity in identifying patterns. As well, this analysis draws on Weber's notion of "verstehen," or "interpretive understanding". As Gann explains [23], verstehen is about grasping the subjective meaning of social action. Our project will apply this interpretive lens to institutional texts – mandate statements, internal memos, media releases – in order to identify not only what was *said*, but also what was *meant* – therefore supplementing the environmental scan with a richer understanding of how mandates were discursively justified to render dissent invisible or illegitimate.

Finally, the application of Weberian concepts – bureaucracy, ideal types, and verstehen – framing the second phase of the project will be informed by Brian Martin's typology of dissent suppression [24], which will help to document how dissenting HCWs were not only excluded in practice, but also delegitimized through tactics such as devaluation, reinterpretation, and institutional silence. This typology will allow the study to go beyond documenting policy timelines and examine how authority, legitimacy, and exclusion were constructed in real time. By analyzing the treatment of dissenting HCWs through the lens of this typology, the project aims to document whether and how such forms of suppression operated in institutional settings during Covid.

### Mapping theoretical frameworks to study phases

The table below presents the alignment between the study's theoretical frameworks and the two main phases of research. Each framework will contribute a distinct analytical perspective to guide data collection, categorization, and interpretation.

| Theoretical Framework | Application | Study Phase |
|---|---|---|
| Max Weber (Ideal Types, Verstehen) | Will be used to construct heuristic ideal types of mandate implementation and to guide interpretive understanding of institutional variation and rationale in policy design. | Phase 1 |
| Carol Bacchi (WPR – 'What's the Problem Represented to Be?') | Will be applied to the analysis of how hospital mandates framed the "problem" of dissident HCWs and to support the discursive deconstruction of mandate justifications. | Phase 2 |
| Brian Martin (Suppression of Dissent Framework) | Will be used to identify mechanisms – discursive, procedural, or reputational – through which dissent or alternative views were marginalized or suppressed. | Phase 2 |

## Aim and objectives

The overall aim of this planned study is to document the implementation of Covid-19 vaccine mandates in medical establishments in the province of Ontario, Canada, understand how authority, legitimacy, and exclusion were constructed in real

time, and elaborate on the scientific, political and ethical implications of these processes. The following research questions will guide the study:

1) How and when were Covid-19 vaccine mandates implemented across medical institutions in Ontario?

2) What were the institutional rationales for vaccine mandates, and how were alternatives to these mandates framed, promoted, tolerated, or suppressed?

## Methods

### Research design

The design of this study consists of two sequential and interrelated parts: (1) an environmental scan of institutional vaccine mandate policies in Ontario medical establishments, and (2) a critical interpretive analysis of these policies using document analysis and the theoretical lenses and tools described earlier. Below we describe the two phases:

- *Phase 1*: this *descriptive* phase will involve an environmental scan, a method developed for the business sector [25] yet also used in other sectors – such as education [26] and increasingly health [27] – to identify and document policy formulation, development and implementation patterns across institutions. The scan will focus on Ontario medical establishments and publicly-available documents will be extracted from institutional websites, media releases, internal memoranda when accessible, and communications from provincial public health authorities – especially Directive #6 issued by Ontario's Chief Medical Officer of Health [28]. A structured table will track: 1) the timeline of mandate implementation, duration, reversals or modifications; 2) alternatives to vaccination permitted; 3) references to scientific justifications; and 4) potential externals influences and responses to policies in the health sector (e.g., media articles; press releases).

- *Phase 2*: this *interpretive* phase will apply document analysis and critical policy analysis to examine how institutions framed their decisions and suppressed dissent. This analysis will draw from Carol Bacchi's WPR framework, which asks what "problem" a policy presupposes to solve, and how such "problem representations" structure the boundaries of available responses. The study will also draw on Brian Martin's typology of the suppression of dissent in science, which includes strategies such as devaluation, reinterpretation, intimidation, and reliance on official channels. These frameworks will guide the examination of how dissenting HCWs were portrayed and managed in institutional discourse, with a focus not on directly identifying or measuring dissent events, but on how institutions described, problematized, and justified responses to dissent or noncompliance. In particular, the study will explore how mandates were framed as unassailable "evidence-based" imperatives and how alternative perspectives were represented or constrained.

Each phase will contribute differently but complementarily to answering the overarching research questions, as they will jointly generate both a *descriptive* mapping of vaccine mandate policy trajectories and a *theoretically informed analysis* of institutional discourse. This dual structure aims to produce a layered understanding of how mandates were justified, operationalized, and contested in the Ontario healthcare sector. The findings are also expected to inform broader discussions about the use of public health authority in institutional decision-making and its implications for professional autonomy, scientific debate, and democratic accountability.

### Data access, selection, and inclusion/exclusion criteria

- *Phase 1:* Hospitals for analysis will be selected based on data from the Canadian Broadcasting Corporation (CBC) Rate My Hospital database, which evaluates approximately 240 acute-care hospitals across Canada, including 153 in Ontario [25]. Each Ontario hospital was classified by the CBC according to (1) institutional type (Teaching, Large Community, Medium Community, or Small Community) and (2) performance grade (A+, A, B, C, or Unrated). These grades were derived from a weighted composition of outcome and quality indicators.

To capture the full diversity of institutional configurations and policy responses, a 20-cell matrix crossing hospital type by grade will be constructed. Hospitals will be placed into their respective cells, and then a purposeful sample will be drawn from each populated cell using maximum variation sampling. This approach aims to document "unique or diverse variations that have emerged in adapting to different conditions" and to identify salient patterns across those variations [29](p.3). For Teaching and Large Community hospitals, institutions with high public visibility (e.g., academic publication output, significant media attention) will be prioritized. The anticipated sample size is 20–25 institutions, balancing representational breadth with feasibility.

The foundational documents for Phase 1 will be each hospital's official policy documents describing Covid-19 vaccination mandates, along with any explanatory or supporting materials such as FAQs, HR bulletins, and public statements or press releases. Inclusion criteria are documents that define, explain, or justify staff vaccination policies; exclusion criteria are documents unrelated to vaccine mandates. To capture contestation and dissent, additional sources will include press releases, media coverage referencing institutional conflict or staff opposition, and legal documents retrieved from the Canadian Legal Information Institute (CanLII) database (e.g., arbitration rulings, grievances, court filings). These materials will allow us to expand perspectives on institutional behavior and identify patterns of justification and response.

- *Phase 2:* This phase will draw upon the same documents collected in Phase 1 but will analyze them using a dissent-suppression framework. We will examine how these documents framed, promoted, tolerated, or suppressed alternatives to vaccination mandates, and how they portrayed dissenting or non-compliant healthcare workers. Where relevant, media reports describing responses to dissent will also be included. No additional primary data will be collected.

### Ethics

The planned study will rely solely on publicly available data that does not engage human subjects. Institutional Review Board approval is waived.

### Timeline

The planned study is expected to be conducted according to the following timeline:

| Month/ Year | Activity |
|---|---|
| April – May/ 2025 | Write protocol; design data collection instruments & submit to MedRxiv & refereed outlets |
| Jun–Sept | Environmental scan phase:<br>• Data collection<br>• Expand literature review |
| Oct – Dec/ 2025 | Environmental scan phase:<br>• Data cleaning, organization & analysis (trends; Weber) |
| Oct – Dec/ 2025 | Environmental scan phase:<br>• Draft manuscript & submit to MedRxiv |
| Jan-March/ 2026 | Critical phase:<br>• Data analysis (Bacchi, Martin) and integration |
| April/ 2026 | Critical phase:<br>• Draft policy manuscript & submit to Preprint.org or similar |
| May/ June 2026 | Critical phase:<br>• Submit to refereed outlets<br>• Further dissemination through single policy brief and lay article |

## Expected outcomes

This planned study will yield descriptive and interpretive insights into how bureaucratic structures shaped mandate enforcement and dissent suppression. Results should inform academic debates on institutional legitimacy, governance, and public health ethics through a variety of strategies of dissemination. In practice, they will culminate in 4 publications:

1. A descriptive report that maps mandate implementation and variation across institutions using environmental scan methodology.

2. A critical analysis of the rationale and discourse surrounding the mandates, drawing on Carol Bacchi's "What is the Problem Represented to Be?" (WPR) framework [30], Martin's typology, critical discourse analysis [31], thematic analysis [32] and document analysis [33].

3. A policy brief that integrates and translates the results of the study for a policy audience.

4. A lay article that integrates and translates the results of the study for the public.

## Statement on reflexivity

This study is informed by the lead investigator's long-standing professional interest in the intersection of policy, governance, and the ethics of healthcare delivery. While the author holds well-developed views about the political and institutional dynamics surrounding Covid-19 mandates, the research design is grounded in a commitment to methodological rigour and epistemic integrity. The aim is not to confirm predetermined assumptions but to systematically document and interpret the evidentiary and discursive basis of institutional decision-making.

Following Finlay [34], reflexivity is approached here not as a means of personal disclosure but as a discipline of methodological transparency and interpretive accountability – meaning not foregrounding the researchers' biographies but rather as reflexive practice deployed to clarify how judgments were made about source selection, interpretive framing, and categorization of mandates. This reflexive orientation allows research like this study to maintain a dual commitment, both to rigorous analysis of institutional policy and to critical self-awareness about how the researcher's position may shape – not predetermine – interpretation.

## Limitations

It may be argued that this project has a built-in bias due to its negative framing of mandated vaccination and is therefore incapable of informing policy. To this we reply that while we have a position on the policy of mandates, the study will not presume their legitimacy or lack thereof, but rather document and explain how they were implemented, justified and contested, and what the tensions identified may reveal about institutional decision-making and professional agency. Critics may also question whether we can expect to gather enough data to support our theoretically informed analysis; they may also question whether the CBC ranking, while offering a consistent starting point for institutional sampling, can truly capture policy diversity or institutional discretion, given the shortcomings of its evaluation criteria, as acknowledged by the CBC itself. We believe that despite these shortcomings – shared by comparable studies, as authors acknowledge [35] – the interpretive phase will be grounded in a diverse and robust corpus of texts – institutional documents and media reports – that should ensure that theoretical insights are empirically anchored, enough to make this study valuable and transferable to other settings and jurisdictions [36].

## Significance

This study should offer timely and broadly relevant insights into how healthcare institutions implemented and justified Covid-19 vaccine mandates, while offering a novel perspective, by treating vaccine mandates not merely as clinical interventions, but as institutional policies subject to political, rhetorical, and ethical contestation. By focusing on Ontario

- Canada's largest province and a leader in healthcare policy – the study will provide a detailed case that can serve as a framework for similar investigations across other provinces. Given that healthcare in Canada is primarily administered and delivered at the provincial level – with federal guidance providing general standards but not operational control [37] – understanding institutional decision-making at the provincial level is essential for evaluating the health system under contested conditions.

The project is also significant for what it can reveal about institutional behaviour in the face of political pressures. While prevailing narratives framed the moment as an emergency [38], responses to vaccine mandates varied widely – not only across jurisdictions, but also between institutions operating under the same provincial directives. These variations raise important questions about how formal organizations enact authority, interpret policy, and manage internal dissent. Max Weber's analytical lens – particularly his insights into bureaucratic rationality and the interpretive meaning of social action – will help to examine how institutions balanced the tensions between competing perceived needs and goals – providing high quality patient care, complying with public health directives, managing internal / sectoral politics, and maintaining internal and external legitimacy. It will also help to understand the tension between structure and agency, i.e., the discretionary space available to decision-makers within bureaucratic systems and the details of the use of this space by medical institutions under the perceived weight of provincial orders.

In addition to these institutional and professional implications, the study may also have relevance for patient outcomes and health care quality. By exploring how healthcare institutions implemented and justified mandates, the research can help illuminate how administrative decisions – shaped by legal authority, rhetorical framing, and organizational discretion – may influence staffing levels, workplace morale, and institutional trust, which in turn can affect the quality, continuity, and equity of care delivered within healthcare systems [39,40], especially under conditions of institutional uncertainty and operational strain.

Further, the project should also contribute to broader debates about how legitimacy is constructed and enforced in public health governance, by documenting not just policy timelines but also rhetorical strategies, decision-making logics, and institutional responses to dissent. In doing so, it should shed light on questions of professional autonomy, evidence-based policy, and the ethics of labour practices in the health sector. Moreover, findings can shed light beyond Canada. For example, in the United Kingdom, with a more centralized health care policy, the government initially announced plans to mandate Covid-19 vaccination for National Health Service workers [41], but in contrast to Canada, it reversed the policy partly due to organized resistance from within the profession [42]. In the United States, where health policy is more decentralized than in Canada, some states – for instance, California – enforced healthcare mandates [43], while others – Texas and Florida – banned them altogether [44]. These divergent approaches highlight the need for grounded, context-sensitive analysis. By documenting the Ontario case in detail, this project should offer conceptual and empirical tools to examine how institutions across jurisdictions justify policies involving coercion, manage opposition, and maintain legitimacy in the name of public health.

## Author contributions

**Conceptualization:** Claudia Chaufan.

**Data curation:** Claudia Chaufan.

**Formal analysis:** Claudia Chaufan.

**Investigation:** Claudia Chaufan.

**Methodology:** Claudia Chaufan.

**Project administration:** Claudia Chaufan.

**Writing – original draft:** Claudia Chaufan.

**Writing – review & editing:** Claudia Chaufan.

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
