## [Decision Letter · Decision Letter 0]

30 Jun 2025

PONE-D-25-25513How did Ontario healthcare institutions implement and legitimize Covid-19 vaccine mandates? A mixed-methods study protocolPLOS ONE

Dear Dr. Chaufan, Thank you for submitting your manuscript to PLOS ONE. After careful consideration, we feel that it has merit but does not fully meet PLOS ONE’s publication criteria as it currently stands. Therefore, we invite you to submit a revised version of the manuscript that addresses the points raised during the review process.

In light of the reviewer's comment, I recommend to incorporate the reviewer's suggestions and address the questions raised by the reviewers.

We look forward to receiving your revised manuscript.

Kind regards,

Mudit Kumar Singh

Academic Editor

PLOS ONE

Journal Requirements:

Reviewers' comments:

Reviewer's Responses to Questions

**Comments to the Author**

1. Does the manuscript provide a valid rationale for the proposed study, with clearly identified and justified research questions?

Reviewer #1: Yes

2. Is the protocol technically sound and planned in a manner that will lead to a meaningful outcome and allow testing the stated hypotheses?

Reviewer #1: Partly

3. Is the methodology feasible and described in sufficient detail to allow the work to be replicable?

Reviewer #1: Yes

4. Have the authors described where all data underlying the findings will be made available when the study is complete?

Reviewer #1: No

5. Is the manuscript presented in an intelligible fashion and written in standard English?

Reviewer #1: Yes

6. Review Comments to the Author

You may also provide optional suggestions and comments to authors that they might find helpful in planning their study.

Reviewer #1: This study is really interesting proposed project, the second research question may need to be reframed as stated now it does not open up the possibility that alternatives were promoted, or at the very least, tolerated in some cases.

(i.e. what were the institutional rationales, and how were alternatives framed, dismissed, or suppressed?)

Further details are needed

-more details on sources for environmental scan would be useful, what specific documents will be included is there any type of inclusion/exclusion criteria?

-what is the selection for the criteria for 20-25 institutions

-where will information for phase 2 come from, how will you find instances of dissent from HCWs? especially if institutions are trying to suppress dissent, will this information be in publicly available documents? would using other publicly available sources also support this phase of the research, please clarify

7. PLOS authors have the option to publish the peer review history of their article (what does this mean?). If published, this will include your full peer review and any attached files.

Reviewer #1: No

---

## [Author Response · Author response to Decision Letter 1]

7 Jul 2025

Point-by-Point Response to Reviewers

1. Reviewer comment: This is a really interesting, proposed project.

Response: Thank you for taking the time to review my work and for your encouraging assessment.

2. Reviewer comment: The second research question may need to be reframed as stated now it does not open up the possibility that alternatives were promoted, or at the very least, tolerated in some cases. (i.e. what were the institutional rationales, and how were alternatives framed, dismissed, or suppressed?)

Response: Thank you for your useful comment. If I understand you correctly you’re asking me to reframe research question 2 to be more open-ended. I have revised question 2 to explicitly allow for documenting tolerance, accommodation, or promotion of alternatives to mandatory vaccination.

3. Reviewer comment: More details on sources for environmental scan would be useful, what specific documents will be included is there any type of inclusion/exclusion criteria?

Response: Thank you for your observation. I have revised the section within methods corresponding to the access, selection, and inclusion exclusion criteria to clarify that the selected institutions are the access point and that the primary sources for the environmental scan will be official hospital policy documents describing COVID-19 vaccination mandates, as well as any explanatory or supporting documents, such as FAQs, HR bulletins, and public statements. I have also clarified that inclusion criteria are documents defining or explaining vaccine policy for staff and exclusion criteria are documents unrelated to vaccine mandates.

4. Reviewer comment: What is the selection for the criteria for 20-25 institutions.

Response: Thank you for your observation. Upon revising the data section (as described earlier) I hope that my discussion of selection criteria is clearer now. In this section I clearly describe the sampling methodology from the initial list of the CBC database.

5. Reviewer comment: Where will information for phase 2 come from?

Response: Thank you for your observation. I have clarified that Phase 2 will analyze the same documents gathered in Phase 1, applying a dissent-framing perspective, with no new data collection

6. Reviewer comment: How will you find instances of dissent from HCWs? especially if institutions are trying to suppress dissent, will this information be in publicly available documents? Would using other publicly available sources also support this phase of the research? Please clarify

Response: Thank you for your observation. Under “research design/phase 2”, I have clarified that the aim is not to identify hidden dissent but to analyze how institutions framed, justified, and responded to dissent in their public or semi-public documentation.

---

## [Decision Letter · Decision Letter 1]

26 Aug 2025

PONE-D-25-25513R1How did Ontario healthcare institutions implement and legitimize Covid-19 vaccine mandates? A mixed-methods study protocolPLOS ONE

Dear Dr. Chaufan,

Thank you for submitting your manuscript to PLOS ONE. After careful consideration, we feel that it has merit but does not fully meet PLOS ONE’s publication criteria as it currently stands. Therefore, we invite you to submit a revised version of the manuscript that addresses the points raised during the review process.

Please address the issues raised by the reviewer specially to bring more clarity to the wider readership.

We look forward to receiving your revised manuscript.

Kind regards,

Mudit Kumar Singh

Academic Editor

PLOS ONE

Journal Requirements:

Reviewers' comments:

Reviewer's Responses to Questions

**Comments to the Author**

1. Does the manuscript provide a valid rationale for the proposed study, with clearly identified and justified research questions?

Reviewer #2: Yes

2. Is the protocol technically sound and planned in a manner that will lead to a meaningful outcome and allow testing the stated hypotheses?

Reviewer #2: Yes

3. Is the methodology feasible and described in sufficient detail to allow the work to be replicable?

Reviewer #2: Yes

4. Have the authors described where all data underlying the findings will be made available when the study is complete?

Reviewer #2: Yes

5. Is the manuscript presented in an intelligible fashion and written in standard English?

Reviewer #2: Yes

6. Review Comments to the Author

You may also provide optional suggestions and comments to authors that they might find helpful in planning their study.

Reviewer #2: Thanks for this interesting study protocol. Here are some recommendations:

1. The study design does not constitute a typical mixed-methods study. Having "mixed-methods" in the title can be confusing for the reader and is best avoided.

2. The background for the proposed work needs to include a discussion on healthcare vaccine mandates that predated and continued past the COVID-19 mandate. The existence of these mandates helped frame the justification for introducing the COVID-19 mandates and needs to recognized in this study.

3. Please clarify for the reader how much latitude individual hospitals had in interpreting the implementation process for the mandate issued by the Ontario Chief Medical Officer of Health. This is vital to understand why variability is to be expected in the different types of hospitals to be included in the study.

4. Consider updating the dates on the timeline table.

5. In the limitations and elsewhere, you mentioned documenting and explaining contestation of the mandates. Re, "document and explain how they were implemented, justified and contested". I remain uncertain as to where documentation of contestations will come from for this study.

7. PLOS authors have the option to publish the peer review history of their article (what does this mean?). If published, this will include your full peer review and any attached files.

Reviewer #2: No

---

## [Author Response · Author response to Decision Letter 2]

30 Aug 2025

Reviewer 2 - optional suggestions and commentaries to the authors

Thanks for this interesting study protocol.

Thank you for your positive assessment of our study protocol. Please find below a point-by-point response to your comments. I appreciate your constructive suggestions and have revised the manuscript accordingly.

Here are some recommendations:

1. The study design does not constitute a typical mixed-methods study. Having "mixed-methods" in the title can be confusing for the reader and is best avoided.

Thank you for this helpful observation. I agree that our design is qualitative and that the term “mixed-methods” could be misleading. I have revised the title and all occurrences in the manuscript to describe the study as a qualitative multi-method protocol, consistent with methodological usage where “multi-method” denotes the combination of two or more qualitative methods (e.g., document analysis, environmental scan of institutional policies/websites, critical policy analysis) without the inclusion of quantitative components (see added reference: Roller, 2021). These changes better reflect the actual design and should avoid confusion for readers.

2. The background for the proposed work needs to include a discussion on healthcare vaccine mandates that predated and continued past the COVID-19 mandate. The existence of these mandates helped frame the justification for introducing the COVID-19 mandates and needs to recognized in this study.

Thank you for this thoughtful suggestion. I have added a short paragraph in the Introduction to contextualize the COVID-19 mandates in relation to prior influenza vaccine policies. While influenza mandates have occasionally been implemented in Canadian healthcare settings, they were typically less coercive and allowed for alternatives such as masking or redeployment. Most importantly, workers who declined influenza vaccination were not vilified, nor were those who complied valorized (see added references: BMJ 2013; Gruben, 2014; Edmond, 2019; Flood, 2021). In contrast, the COVID-19 mandates were accompanied by a dramatic shift in institutional tone and public framing — from heroism to threat — even though Ontario’s Directive 6 permitted multiple, far less intrusive pathways to compliance (e.g., an educational module and testing). This change highlights the need to examine not only policy content but also how it was enacted and justified at the institutional level.

3. Please clarify for the reader how much latitude individual hospitals had in interpreting the implementation process for the mandate issued by the Ontario Chief Medical Officer of Health. This is vital to understand why variability is to be expected in the different types of hospitals to be included in the study.

Thank you for this suggestion. I have clarified this point in the Introduction and throughout Phase 1. As revised, the manuscript now explains that Ontario’s Directive 6 mandated that all covered organizations implement a COVID-19 vaccination policy but granted significant discretion to each institution in how to operationalize it. The Directive permitted multiple compliance options — including full vaccination, medical exemption, and participation in an educational session followed by regular antigen testing. While the Directive formally preserved institutional choice, most hospitals implemented only the most restrictive version. Our study explicitly investigates this pattern of selective implementation, its justification, and its implications. I believe the revised manuscript now clearly addresses your concern.

4. Consider updating the dates on the timeline table.

Thank you for this suggestion. The timeline has been updated to reflect the current stage of the project and align with the revised protocol. The changes are administrative and clarify the sequencing of activities between Phase 1 and Phase 2. No major design or methodological changes have been made. As before, our plan is to begin manuscript preparation and submission upon completion of each phase, rather than wait for the full project to conclude. This reflects our typical workflow, which prioritizes timely dissemination while maintaining continuity across the research program.

5. In the limitations and elsewhere, you mentioned documenting and explaining contestation of the mandates. Re, "document and explain how they were implemented, justified and contested". I remain uncertain as to where documentation of contestations will come from for this study.

Thank you for this important question. As clarified in the revised manuscript (see Introduction and Phase 1 description), we will document contestation through a diverse range of publicly available sources. These include: (1) legal challenges filed by employees or unions (e.g., arbitration rulings, grievances, court filings) retrieved via the Canadian Legal Information Institute (CanLII); (2) media coverage describing institutional conflicts, staff terminations, or episodes of internal dissent; and (3) hospital-generated materials (e.g., HR bulletins, FAQs, press releases) that may indirectly acknowledge contestation — for instance, by outlining consequences for noncompliance or providing justifications for disciplinary measures. While these sources do not capture all forms of dissent, they offer important insight into how hospitals publicly framed and managed noncompliance. These materials will form the basis for the interpretive phase of our study, which focuses on institutional discourse and the representation (or silencing) of contestation.

---

## [Editor Report · Decision Letter 2]

3 Sep 2025

How did Ontario healthcare institutions implement and legitimize Covid-19 vaccine mandates? A qualitative multi‑method study protocol

PONE-D-25-25513R2

Dear Dr. %Chaufan%,

We’re pleased to inform you that your manuscript has been judged scientifically suitable for publication and will be formally accepted for publication once it meets all outstanding technical requirements.

Kind regards,

Mudit Kumar Singh

Academic Editor

PLOS ONE
---

## [Editor Report · Acceptance letter]

PONE-D-25-25513R2

PLOS ONE

Dear Dr. Chaufan,

I'm pleased to inform you that your manuscript has been deemed suitable for publication in PLOS ONE. Congratulations! Your manuscript is now being handed over to our production team.

Kind regards,

on behalf of

Dr. Mudit Kumar Singh

Academic Editor

PLOS ONE